# Cytoskeletal Actin Structure in Osteosarcoma Cells Determines Metastatic Phenotype via Regulating Cell Stiffness, Migration, and Transmigration

**Kouji Kita [1], Kunihiro Asanuma [1,*], Takayuki Okamoto [2], Eiji Kawamoto [3], Koichi Nakamura [1], Tomohito Hagi [1], Tomoki Nakamura [1], Motomu Shimaoka [3] and Akihiro Sudo [1]**

[1] Department of Orthopedic Surgery, Mie University Graduate School of Medicine, 2-174 Edobashi, Tsu 514-8507, Mie, Japan; kkita125@gmail.com (K.K.); k-nakamura@clin.medic.mie-u.ac.jp (K.N.); hagifana@clin.medic.mie-u.ac.jp (T.H.); tomoki66@clin.medic.mie-u.ac.jp (T.N.); a-sudou@clin.medic.mie-u.ac.jp (A.S.)

[2] Department of Pharmacology, Faculty of Medicine, Shimane University, 89-1 Enya-cho, Izumo-shi 693-8501, Shimane, Japan; okamoto@med.shimane-u.ac.jp

[3] Department of Molecular Pathobiology and Cell Adhesion Biology, Mie University Graduate School of Medicine, 2-174 Edobashi, Tsu 514-8507, Mie, Japan; a_2.uk@mac.com (E.K.); shimaoka@doc.medic.mie-u.ac.jp (M.S.)

* Correspondence: kasanum@gmail.com; Tel.: +81-59-231-5022; Fax: +81-59-231-5211

**Abstract:** Osteosarcoma is the most common primary malignant bone tumor. The cause of death due to osteosarcoma is typically a consequence of metastasis to the lung. Controlling metastasis leads to improved prognosis for osteosarcoma patients. The cell stiffness of several tumor types is involved in metastatic potential; however, it is unclear whether the metastatic potential of osteosarcoma depends on cell stiffness. In this study, we analyzed the cell stiffness of the low metastatic Dunn cell line and its highly metastatic LM8 subline, and compared actin organization, cell proliferation, and metastasis. Actin cytoskeleton, polymerization, stiffness, and other cellular properties were analyzed. The organization of the actin cytoskeleton was evaluated by staining F-actin with Alexa Fluor 488 phalloidin. Cell stiffness was measured using Atomic Force Microscopy (AFM). Cell proliferation, migration, invasion, and adhesion were also evaluated. All experiments were performed using mouse osteosarcoma cell lines cultured in the absence and presence of cytochalasin. In LM8 cells, actin polymerization was strongly suppressed and actin levels were significantly lower than in Dunn cells. Stiffness evaluation revealed that LM8 cells were significantly softer than Dunn. Young's modulus images showed more rigid fibrillar structures were present in Dunn cells than in LM8 cells. LM8 cells also exhibited a significantly higher proliferation. The migration and invasion potential were also higher in LM8 cells, whereas the adhesion potential was higher in Dunn cells. The administration of cytochalasin resulted in actin filament fragmentation and decreased actin staining intensity and cell stiffness in both LM8 and Dunn cells. Cells with high metastatic potential exhibited lower actin levels and cell stiffness than cells with low metastatic potential. The metastatic phenotype is highly correlated to actin status and cell stiffness in osteosarcoma cells. These results suggest that evaluation of actin dynamics and cell stiffness is an important quantitative diagnostic parameter for predicting metastatic potential. We believe that these parameters represent new reliable quantitative indicators that can facilitate the development of new drugs against metastasis.

**Keywords:** metastasis; cell stiffness; atomic force microscopy; osteosarcoma cell line; actin cytoskeleton

## 1. Introduction

Osteosarcoma is the most common primary malignant tumor of bone. Lung metastasis occurs in approximately 50% of patients with osteosarcoma and is a major cause of death. [1,2]. Since metastases of osteosarcoma are detectable in only 10 to 20% of patients at diagnosis [3], the prediction of metastasis holds promise for improving prognosis.

Diagnosis, staging, and surgical management of osteosarcoma remain focused on our anatomical understanding of the disease. Classical prognostic factors among patients with osteosarcoma include tumor size, primary site, histological grade and the histological response to preoperative chemotherapy. Numerous studies have reported factors that can be linked to poor outcomes, such as elevated serum alkaline phosphatase levels, serum lactate dehydrogenase, and alkaline phosphatase [4–7]. The current comprehensive understanding of the molecular pathogenesis of osteosarcoma expands therapeutic potential and/or diagnostic targets that are being developed [8–10]. Lung metastasis, despite successful surgical resection, is critical for the prognosis of osteosarcoma; however, the determinant factors of metastasis remain to be identified. The ability to predict and control metastasis is expected to improve prognosis for osteosarcoma patients.

Metastasis is a complex process in which cancer cells spread from a primary site to other organs. Metastasis to distal organs involves cancer cells orchestrating the secretion of proteases, as well as alterations in adhesion receptors, and changes in cell morphology and migratory properties. Furthermore, recent studies have shown that the physical properties of cancer cells are implicated in metastatic processes. High deformability is thought to be an important property of metastatic cancer cells. Cellular deformability is strongly correlated with cell stiffness, which is defined as Young's modulus [2]. Cell stiffness is controlled by the distribution and organization of the actin network [11,12]. Additionally, actin plays an important role in modulating cell shape, motility, invasion, and migration. Cells transforming from a non-malignant to a malignant phenotype exhibit an altered cytoskeletal structure, namely changing from an organized to an irregular network associated with a subsequent reduction in stiffness [11,12]. Docheva et al. reported that human osteoblasts (hOBs) showed the presence of large focal adhesions and bulky actin stress fibers; conversely, human osteosarcoma cells (MG63) showed small, sparse focal adhesions and thin and less abundant stress fibers. Evaluation of stiffness revealed that hOBs were stiffer than MG63 cells [13]. Xu et al. reported ovarian cancer cells (HEY) were softer than non-malignant ovarian epithelial cells. Moreover, a highly invasive phenotype (HEY A8) was found to be much softer than non-malignant ovarian or HEY cells [14].

These reports suggest that cell stiffness is involved in malignancy and metastatic potential. Thus, we hypothesized that cell stiffness is related to metastasis in osteosarcoma. To elucidate the correlation between cell stiffness and metastatic potential in osteosarcoma, we compared actin structures, polymerization, stiffness, and other cellular properties of two osteosarcoma cell lines. The parental mouse osteosarcoma cell line (Dunn) has a low metastatic phenotype and LM8 cells have a highly metastatic phenotype. LM8 cells have the unique feature of undergoing spontaneous lung metastasis from subcutaneous transplantation sites without intravenous cell injection or solid tumor resection. This metastatic phenotype is thought to be a closer representation of clinical metastasis in humans than other metastatic models in mice.

## 2. Materials and Methods

### 2.1. Cell Culture

The highly metastatic LM8 cell line was derived from the Dunn cell line through eight repeated cycles of the procedure described by Poste and Fidler [15]. The mouse LM8 cell line (Suita, Osaka, Japan, December 2014) was kindly provided by Osaka University [16]. The Dunn cells were maintained in Minimum Essential Medium (MEM: Gibco BRL, Grand Island, NY, USA), while the LM8 cells were maintained in Dulbecco's modified Eagle's medium (DMEM: Gibco BRL, Grand Island, NY, USA) supplemented with 10% bovine serum albumin (BSA). Cells were cultured at 37 °C in an incubator under 5% $CO_2$. Cytochalasin inhibits actin polymerization, presumably by binding with high affinity to the barbed ends of actin filaments (F-actin) and preventing the addition of monomers (G-actin) to these sites [17]. Thus cytochalasin reduces cell stiffness by disrupting the actin cytoskeleton [18,19]. In each experiment, cytochalasin was administered at a concentration of 2 µM.

## 2.2. Evaluation of Metastatic Potential In Vivo

We evaluated the metastatic potential of Dunn and LM8 in vivo. Mice were randomly divided into two groups ($n = 7$/group). After suspending Dunn and LM8 cells in phosphate-buffered saline (PBS), 200 μL of the cell suspension ($1 \times 10^7$ cells) was injected into the dorsal subcutaneous of C3H/He female mice. The mice were sacrificed at 5 weeks after injection and the lungs was carefully excised. The excised lungs were fixed with formalin, embedded in paraffin, sectioned and stained with hematoxylin and eosin for histological observation. The lung metastasis was measured by counting the metastatic node using hematoxylin and eosin staining.

## 2.3. Fluorescence Staining and Analysis of F-Actin

We imaged the fluorescently labeled F-actin network using microscopy. Cells were grown on a glass coverslip to a density of $3.0 \times 10^4$ cells/mL with 500 μL cell culture media. The cells were incubated at 37 °C for 48 h and the actin cytoskeleton stained with Alexa 488-conjugated phalloidin. The cells were fixed with 4% paraformaldehyde containing 0.1% Triton-X for 30 min. The cells were blocked with 1% BSA for 30 min at room temperature and subsequently incubated for 20 min with Alexa Fluor 488 phalloidin (1:40; Thermo, Tokyo, Japan, Molecular Probes) in 1% BSA. All images were captured using a microscope (Olympus, Tokyo, Japan) at an excitation wavelength of 488-nm with a 100-ms exposure time. Schaub et al. reported that the fluorescence intensity of phalloidin-stained images is proportional to the density of actin filaments and that variations in staining intensity reflect variations in actin density [18]. We used Image J software (NIH Image, Bethesda, MD, USA) to measure the mean intensity of F-actin staining [18]. To measure intensity variations, we used the ratio of standard deviation (SD) of intensity to the mean intensity of the image. Contrast is expected to be proportional to the inverse of the square root of the concentration, which was calculated using a previously reported formula [20].

## 2.4. Measurement of Cell Stiffness

In this study, atomic force microscopy (AFM) was performed using a NanoWizard 3 AFM system (JPK Instruments AG, Berlin, Germany) with a cantilever and a tetrahedral type probe (BL-AC40TS-C2; Olympus, Tokyo, Japan). AFM was used to measure the mechanical properties of the cells. The plot of the measured force against the deformation of the sample (force-distance curve) provides information on the material properties, such as Young's modulus. Luo et al. reported that the dissolution of the actin cytoskeleton led to a significant decrease in Young's modulus. Therefore, in this study, Young's modulus was used as an index of cell stiffness [21].

The cells were grown on glass bottomed culture dishes at a density of $3.0 \times 10^4$ cells/mL, incubated at 37 °C for 48 h. Thereafter, measurements were conducted in cell culture media at room temperature. All force curves and scanning field images (10 μm × 10 μm) were recorded at a resolution of 128 × 128 pixels in Quantitative imaging (QI) mode at 37 °C.

Measurement of cell stiffness using AFM was performed as previously described [22]. The data were processed by curve-fitting with the Hertz contact model using JPK data processing software. The geometric mean of Young's modulus was calculated from the acquired Young's modulus at each point of the cell for a given condition.

## 2.5. Cell Proliferation Assay

Dunn and LM8 cells were seeded in 96-well plates in a 100 μL volume at a density of $3.0 \times 10^4$ cells/mL, incubated at 37 °C. After 24, 48, and 72 h, cell proliferation was measured using the CellTiter 96$^{TM}$ AQueous Non-radioactive Cell Proliferation Assay (Promega, Mannheim, Germany).

## 2.6. Migration Assay

Cell migration was assessed using a wound-healing assay. The cells were cultured to confluence and then scratched with a 10-μL pipette tip and imaged. Images were acquired

at 0, 24, 48, and 72 h post-wounding with a microscope (Olympus, Tokyo, Japan) ($\times$200) and analyzed with Image J software. The rate of cellular migration was determined as cells moving from the intact zones into the scratched region at the different time points. The evaluation method sets the cell-free area at 0 h to 100% and determines the rate of reduction in the cell-free area.

### 2.7. Invasion Assay

Cellular invasion was analyzed using Matrigel invasion chambers (BD Pharmingen, Corston, UK) in 24-well plates. Dunn and LM8 cells were seeded in the upper chamber in a 500 μL volume at a density of $5.0 \times 10^4$ cells/mL. The lower chamber was filled with medium. The cells were incubated for 48 h at 37 °C in an incubator under 5% $CO_2$. The invading cells migrate in a vertical direction through the pores of the membrane of the upper chamber into the lower chamber.

After rinsing with PBS, the filter was fixed, and hematoxylin and eosin staining was performed. Cellular invasion was determined by counting the stained cells on the bottom surface of the upper chamber following imaging of three random fields of the invading cells with a microscope ($\times$200). The number of cells on the bottom surface of the upper chamber was compared between groups [23].

### 2.8. Adhesion Assay

Cell adhesion was analyzed using 96-well V-bottom plates. Centrifugal force is applied to separate adherent from nonadherent cells. The force produced by the centrifugation step results in the accumulation of free or loosely attached cells in the tip of the V-shaped wells. V-bottom plates were blocked with 1% sterile BSA in PBS buffer (200 μL) for 1 h at 37 °C in an incubator. The cells were labeled with 2′,7′-bis(2-carboxyethyl)-5,6-carboxyfluorescein (BCECF) 1 μL/mL for 30 min at 37 °C. The cells were diluted in HBS and BCECF-labeled cells were dispensed in each well in 100 μL at a density of $9.0 \times 10^4$ cells/mL, and then centrifuged at 280 g (1200 rpm) for 5 min using a swinging bucket rotor (EX-125; Tomy Seiko Co., Ltd., Osaka, Japan). Nonadherent cells accumulated at the nadir of the V-bottom wells and were quantified with 2030 ARVO X-2 Multilabel Reader (PerkinElmer Japan Co. Ltd., Kanagawa, Japan) [24].

### 2.9. Statistical Analyses

All data are presented as mean $\pm$ SD. Statistical analysis of differences in the mean values of Young's modulus was conducted using nonparametric analysis with the Wilcoxon-Mann-Whitney test. In the evaluation of stiffness, the number of samples in each group was 7 and the experiment was performed three times. For all statistical tests, a *p* value < 0.05 was considered significant.

Statistical analyses were performed using the EZR graphical user interface [25].

## 3. Results

### 3.1. LM8 Had Higher Metastatic Potential Than Dunn

In this study, we evaluated metastatic potential by counting the number of lung metastatic nodules between Dunn and LM8. In hematoxylin and eosin staining, the number of metastatic foci of LM8 in the lung per mouse was significantly higher than Dunn. This indicated that LM8 had high metastatic potential. (Figure 1a,b)

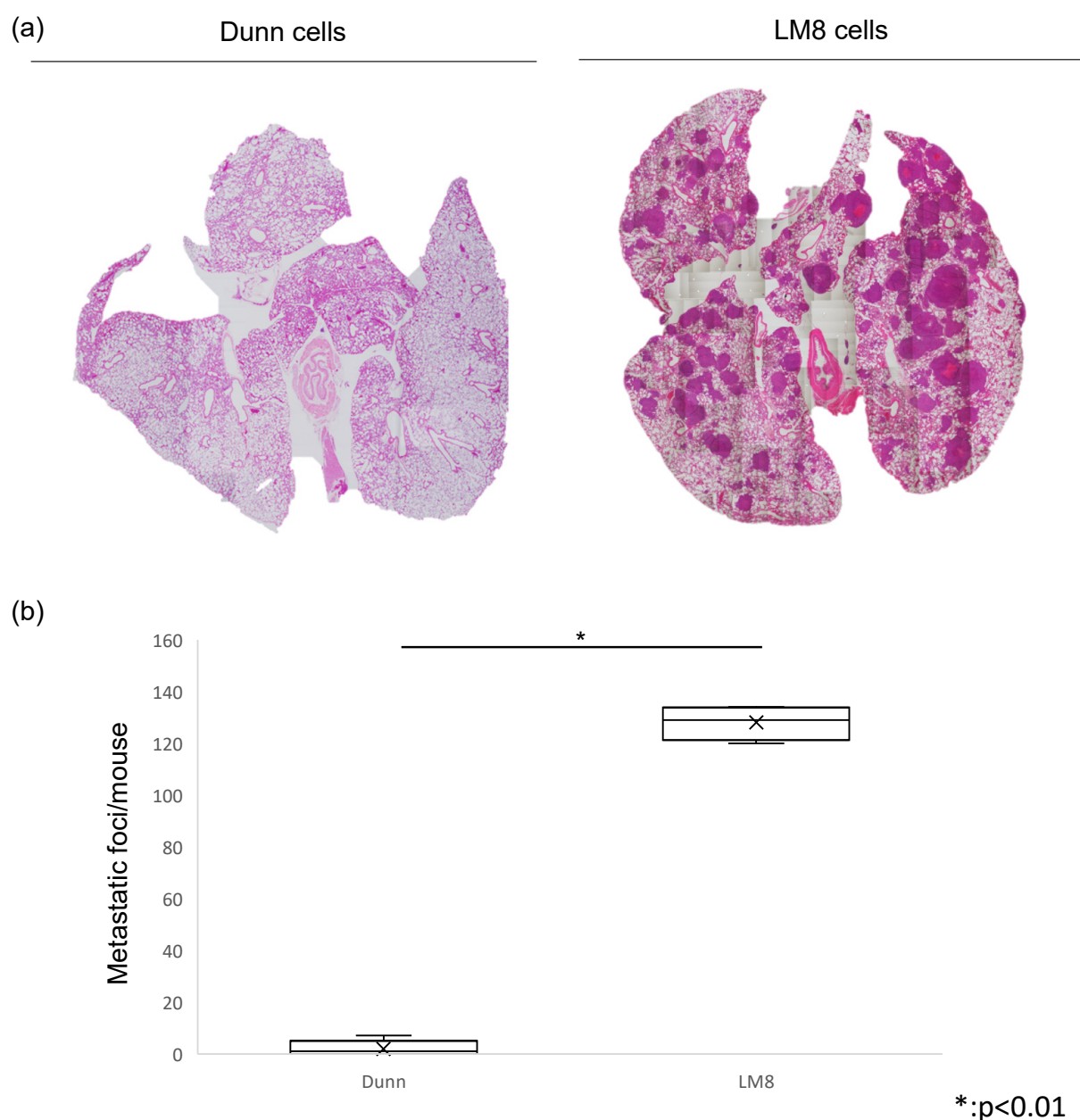

**Figure 1.** Evaluation of lung metastases between Dunn and LM8. (**a**) Hematoxylin and eosin staining was performed (**Left**: Dunn, **Right**: LM8). The metastatic foci was counted. (**b**) The number of metastatic foci of LM8 in the lung was significantly higher than Dunn (* $p < 0.01$).

### 3.2. Actin Polymerization, Intensity, and Stiffness Was Reduced in the Highly Metastatic LM8 Subline

Actin polymerization, distribution, and organization were investigated. Fluorescence imaging of F-actin indicated that actin polymerization was strongly suppressed in LM8 cells. The actin filaments in LM8 cells were shorter, less organized, and randomly oriented. Conversely, the actin filaments in Dunn cells were distributed throughout the cell body and aligned along the long axis of the cells. (Figure 2a) Differences in actin-staining intensity were evaluated by Image J software, and indicated that actin levels in LM8 cells was significantly lower than in Dunn cells. (Figure 2a,b)

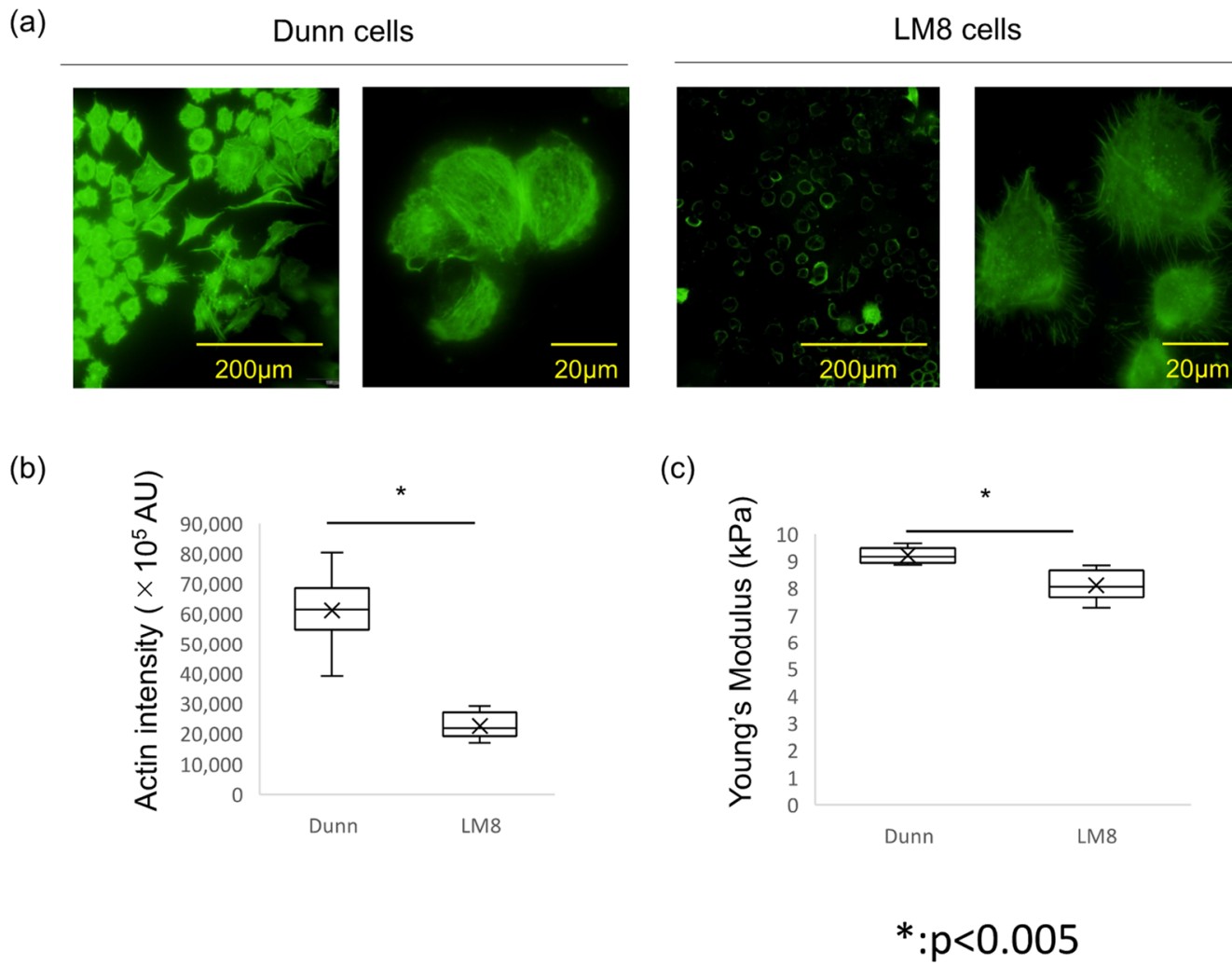

**Figure 2.** Actin polymerization, intensity, and stiffness between Dunn and LM8. (**a**) Actin cytoskeleton was stained with Alexa 488-conjugated phalloidin. **Left**: Dunn, **Right**: LM8. (**b**) Actin staining intensity was evaluated by arbitrary units (AU) using Image J software. * $p < 0.005$. (**c**) Young's modulus was measured by AFM. Mean $\pm$ SD, Dunn: 9.20 $\pm$ 0.27 kPa. LM8: 8.11 $\pm$ 0.51 kPa. * $p < 0.005$.

Furthermore, we used AFM to measure cell stiffness. Seven Dunn and LM8 cells each were measured, with 16,384 points/cell. The geometric means of the Young's modulus were compared. The mean stiffness of LM8 and Dunn cells was 8.11 $\pm$ 0.51 kPa and 9.20 $\pm$ 0.27 kPa, respectively, with LM8 cells significantly softer than Dunn cells ($p < 0.005$). (Figure 2c) Young's modulus images were reconstructed and showed that more rigid fibrillar structures were observed in Dunn cells compared to LM8 cells. (Figure 3a,b) These results indicated that cellular stiffness is highly proportional to actin levels. Moreover, there is a strong relationship between actin levels, cell stiffness, and metastatic potential.

(a)

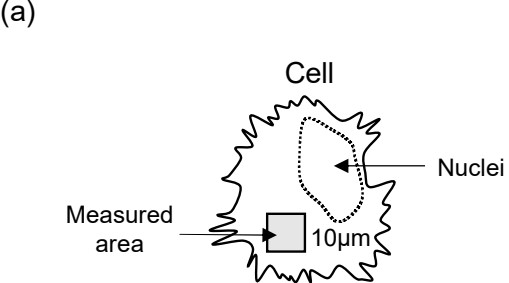

(b)

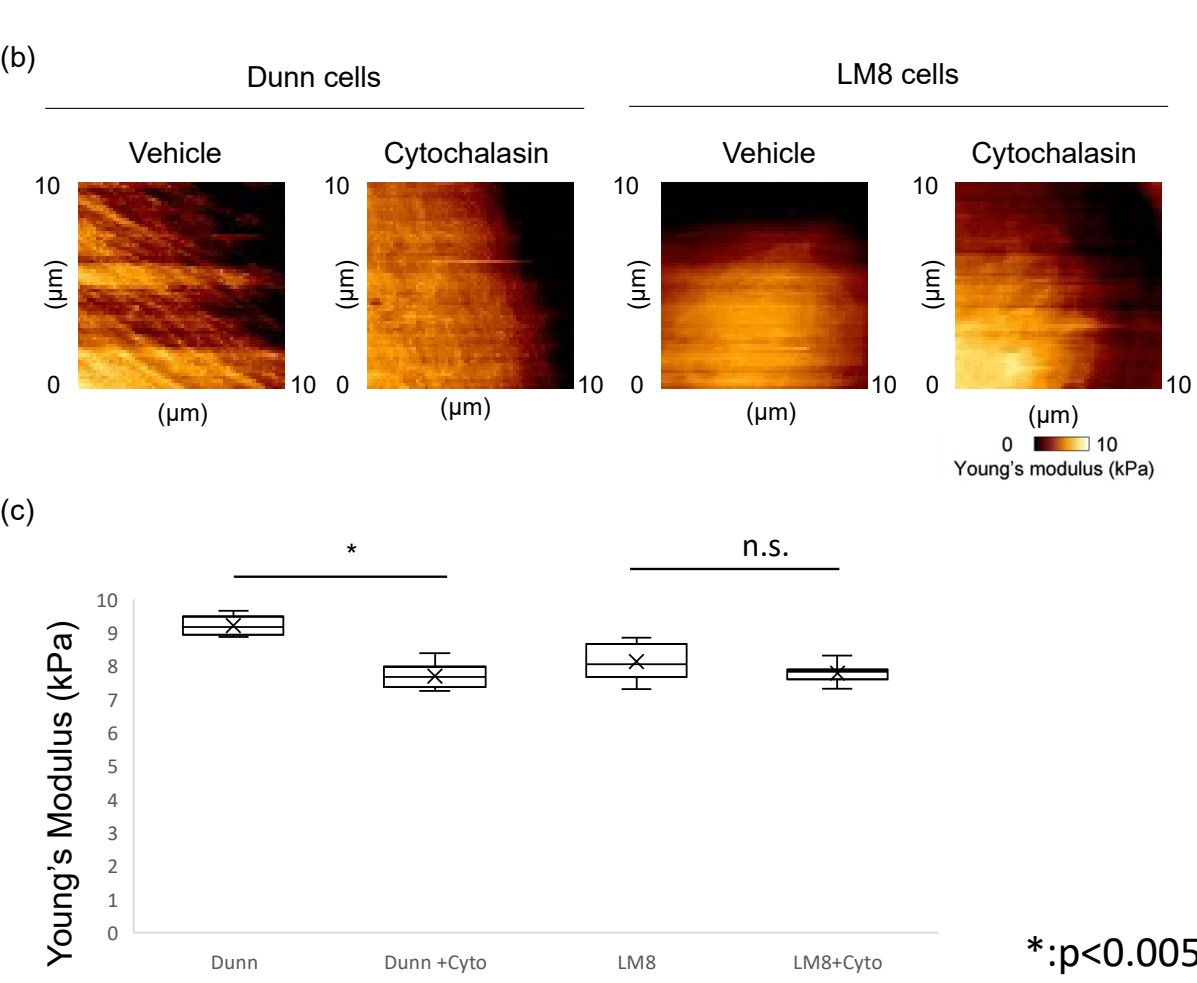

**Figure 3.** Change of Young's modulus by cytochalasin. (**a**) Measured area by AFM was indicated in the scheme. (**b**) Young's modulus images in Dunn (**left**) and LM8 (**right**) with or without cytochalasin were shown. (**c**) Young's modulus was compared between Dunn and LM8 with or without cytochalasin. Dunn (−: 9.20 ± 0.27 kPa, +: 7.69 ± 0.35 kPa, $p < 0.005$), LM8 (−: 8.11 ± 0.51 kPa, +: 7.77 ± 0.26 kPa, * $p < 0.005$, n.s.: not significant).

### 3.3. Actin Depolymerization by Cytochalasin Affects Actin Levels and Cell Stiffness

Treatment with the actin polymerization inhibitor cytochalasin was used to induce changes in actin structure and examine the role of actin in determining cell stiffness. Treatment with cytochalasin resulted in shortening and fragmentation of actin filaments in both LM8 and Dunn cells (Figure 4a). Additionally, actin staining intensity was significantly reduced in the presence of cytochalasin in both LM8 ($p < 0.005$) and Dunn ($p < 0.01$) cells (Figure 4b).

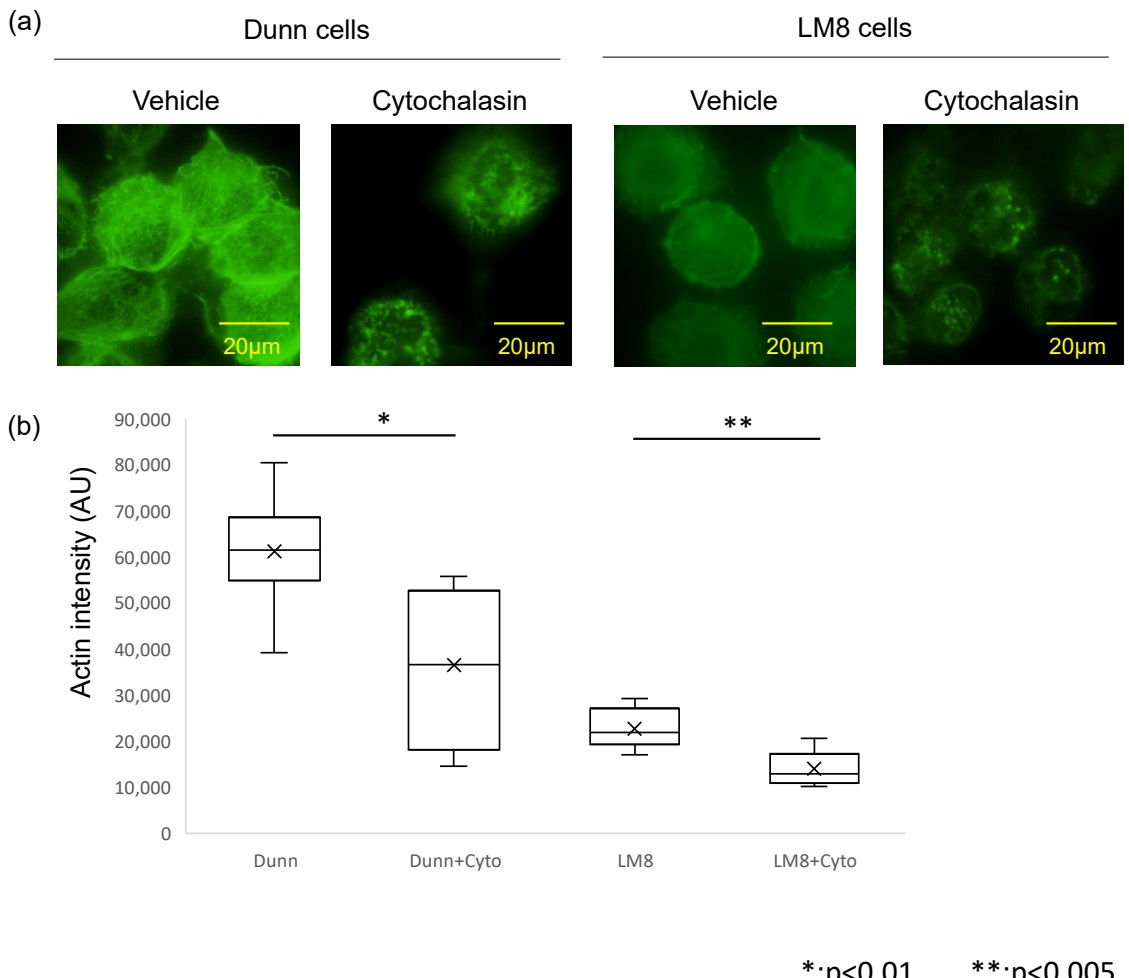

**Figure 4.** Actin polymerization and intensity with or without cytochalasin. (**a**) Actin filaments were visualized and compared between Dunn (**left**) and LM8 (**right**) with or without cytochalasin. (**b**) Actin intensity was evaluated between Dunn and LM8 with or without cytochalasin.

While cell stiffness decreased from $8.11 \pm 0.51$ kPa to $7.77 \pm 0.26$ kPa in LM8 cells, a statistically significant difference was not observed. In contrast, cell stiffness was significantly decreased ($p < 0.005$) from $9.20 \pm 0.27$ kPa to $7.69 \pm 0.35$ kPa in Dunn cells (Figure 3c). Furthermore, cytochalasin impaired rigid fibrillar structures and cell stiffness of LM8 and Dunn cells (Figure 3b).

Cytochalasin inhibits actin polymerization and reduces the stiffness of LM8 and Dunn cells to similar levels of hardness. In previous reports, other actin polymerization inhibitors such as LeftyA and Latrunculin A reduced cell stiffness [26–29]. These indicate that actin polymerization is strongly involved in determining cell stiffness. Furthermore, analysis of actin staining intensity produced results similar to those observed for cell stiffness. These results suggest that actin changes could account for the observed differences in cell stiffness in the examined mouse osteosarcoma cell lines.

### 3.4. Evaluation of LM8 Cell Metastatic Potential

An analysis of functional differences between LM8 and Dunn cells was performed, including cellular proliferation (Figure 5a), migration (Figure 5b), invasion (Figure 5c), and adhesion (Figure 5d). Cellular proliferation and migration were determined after 24, 48, and 72 h in culture. LM8 cells exhibited significantly higher proliferation after 48 h ($p < 0.005$) than Dunn cells. Moreover, LM8 cells exhibited significantly higher invasion and

significantly lower adhesion than Dunn cells. Thus, the migration and invasion potential were higher in LM8 cells, while the adhesion potential was higher in Dunn cells ($p < 0.005$).

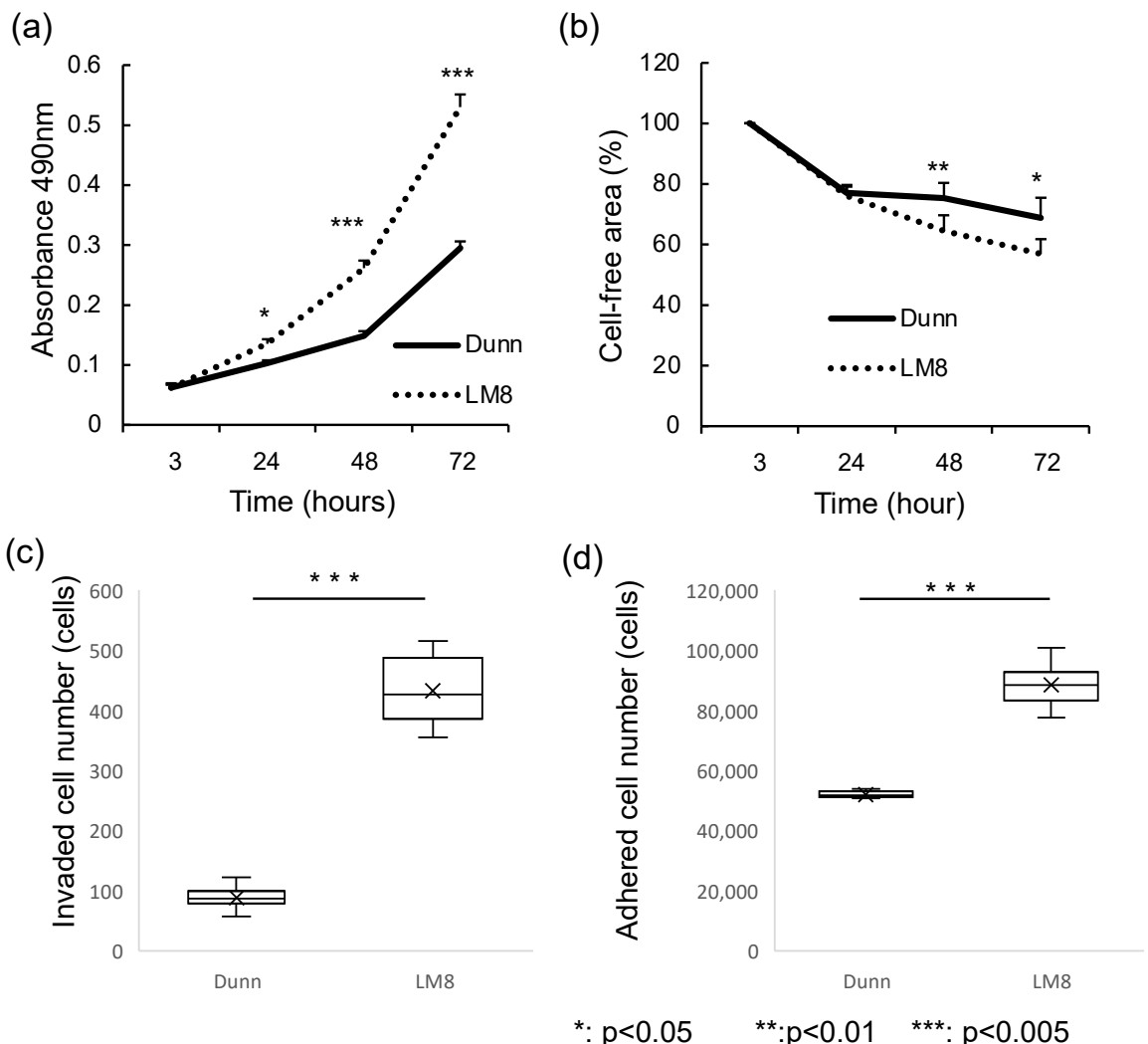

**Figure 5.** Functional analysis of LM8 and Dunn cells. (**a**) Cell proliferation and (**b**) migration were measured after 3, 24, 48, and 72 h in culture. (**c**) Invasion activity were calculated after 48 h treatment. (**d**) Adhesion activity was evaluated after 1 h treatment.

## 4. Discussion

Metastasis by malignant cells is a common and critical phenomenon that has been studied extensively; however, the underlying mechanisms involved remain to be fully elucidated. Motility, migration, invasion, and proliferation are barometers of metastatic potential in vitro. In recent years, attention has been focused on the relationship between cell deformability and metastatic potential as a new barometer of metastatic potential.

Cell deformability, flexibility, and compliance are important factors enabling the passage of tumor cells through the extracellular matrix [30]. Cell deformability had strong correlation with cell stiffness defined as Young's modulus [2,31]. Young's modulus is regarded as a biomarker of cancer cell metastasis [21]. Thus, the present study used two osteosarcoma cell lines to elucidate the correlation between cell stiffness and metastatic phenotype in osteosarcoma. LM8 cells are a highly metastatic derivative of Dunn cells that were generated by an in vivo selection method [16]. LM8 cells have a unique phenotype in that they form lung metastases following subcutaneous implantation, which is similar to

clinical observations of osteosarcoma metastasis. Thus, the mechanism of metastasis could be elucidated by comparing these two cell lines [32,33].

First, we evaluated actin staining intensity to elucidate the relationship between metastasis and actin filaments. Cell deformability is regulated by the actin cytoskeleton, and Guck et al. reported that the increased deformability of cancer cells is accompanied by a reduction in the amount of F-actin [11]. It was found that cell stiffness was proportional to actin staining intensity, and that changes in the actin cytoskeleton alter cell deformability. These results indicate that the actin cytoskeleton may affect cell stiffness. Fluorescence imaging of F-actin morphology revealed that actin filaments in Dunn cells were distributed throughout the cell body and aligned along the long axis of the cells. Conversely, the actin filaments in LM8 cells were shorter, less organized, and oriented randomly. Quantitative analysis indicated that actin levels in LM8 cells were significantly lower than in Dunn cells. These results suggest that F-actin levels are related to cell stiffness and metastatic potential.

Secondly, we evaluated Young's modulus using AFM to elucidate the relationship between cell stiffness and metastasis. Swaminathan et al. reported that cancer cells with the highest migratory and invasive potential are five times less stiff than cells with the lowest migratory and invasive potential [34]. The elastic modulus represents the deformability or compliance of individual cells [31], and cells with lower metastatic potential had significantly higher elastic modulus values (were less deformable) than cells with higher metastatic potential [35]. In this study, the mean stiffness of LM8 cells ($8.11 \pm 0.51$ kPa) was significantly lower than Dunn cells ($9.20 \pm 0.27$ kPa) ($p < 0.01$). This indicates that cell stiffness is strongly associated with metastatic potential. Given this, we successfully demonstrated that the metastatic phenotype is highly correlated with actin levels and cell stiffness in osteosarcoma cells.

Based on our results, we propose the hypothesis that increasing cell stiffness can inhibit metastasis in osteosarcoma. Cancer cell malignancy is typically evaluated by in vitro assays of cell proliferation, migration, invasion, and adhesion. Cell stiffness is a new factor for evaluating cell malignancy or metastasis and can be used as a new parameter for the development of new anti-metastatic drugs. Screening for drugs that increase cell stiffness is a new approach to drug development that has potential for identifying new therapeutic agents targeting metastasis. Measuring stiffness by AFM has the advantage of being able to quantitatively determine cell stiffness of individual living cells in physiological environments. However, the disadvantage is that measurements require a long time and can only be performed on a limited number of cells. In this study, we successfully showed a correlation between actin levels and cell stiffness in both the absence and presence of cytochalasin. Measurement of actin staining intensity is easier and less time-consuming than measuring Young's modulus, and can be used as a first step in identifying drug candidates that potentially alter Young's modulus.

In conclusion, we demonstrated that metastatic phenotype is highly correlated to actin levels and cell stiffness in Dunn and LM8 osteosarcoma cells. Actin levels and cell stiffness may be used as new parameters to determine metastatic potential and as quantitative indicators in the development of new drug targeting metastasis. This needs further study.

**Author Contributions:** Main manuscript, K.K.; Conceptualization, K.K. and K.A.; Experiments, K.N., E.K. and K.K.; Data analysis, T.H. and T.N.; M.S., T.O. and K.A. critically reviewed the manuscript; A.S. supervised this study. All authors have read and agreed to the published version of the manuscript.

**Funding:** This research received no external funding.

**Institutional Review Board Statement:** The study was approved by the Ethics Committee of Mie University (protocol code 2019-35 and the date of approval 6 January 2020).

**Informed Consent Statement:** Not applicable.

**Data Availability Statement:** Not applicable.

**Conflicts of Interest:** The authors declare that they have no conflict of interest.

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
