# Peer review of "Cytoskeletal Actin Structure in Osteosarcoma Cells Determines Metastatic Phenotype via Regulating Cell Stiffness, Migration, and Transmigration"

_cimb, doi:10.3390/cimb43030089_

Round 1

Reviewer 1 Report

This paper titled 'Cytoskeletal actin structure in osteosarcoma cells determines metastatic phenotype via regulating cell stiffness, migration, and transmigration focuses on differences between two osteosarcoma cell lines in terms of their metastatic potential. 

-As a general rule, citation brackets should come at the end of the sentences before the 'full stop' punctuation mark. 

-In general, bar graphs can be re-evaluated and converted to a version showing single data points. 

-This paper only includes a pair of osteosarcoma cells to test the hypothesis. To make a solid conclusion, all experiments should be repeated with another osteosarcoma cell line pair (one should be more metastatic than the other) and results should be evaluated. Otherwise, these results could be considered as an observation only for these two osteosarcoma pairs.

-Line 195: It has been estimated that LM8 cells are more metastatic than Dunn cells in general, and the paper is constructed on this assumption, but there is not any biological data to support this assumption. Although these two cells lines are very well-established models for these experiments, there needs to be data showing their metastatic potential at the beginning of the paper.

-Line 203: This experiment can be repeated with another actin polymerization inhibitor just to confirm if the results really depend on the actin polymerization.

-Line 276: This is an overstatement. This hypothesis can be specific for bone metastasis, but I do not think it is valid for all other cancer types.

Reviewer 2 Report

The manuscript by Kita et al. describes the importance of cytoskeletal actin in the malignant phenotype of osteosarcoma cells. Although the obtained results is with great interest several points need to be addressed.

  • My major concern are the conditions where the comparison between Dunn cells and LM8 was performed. Indeed, although these cells are from the same origin they are cultured in different media. I m wondering whether the differences in these conditions may also interfere with the malignant phenotype of these cells. For a better comparison the experiments should be done (at least the major such as the invasion assay) using the same culture media for both cells. Other option and it will be the best option is to use additional cell lines for validation.
  • In the page 7 : (3. Evaluation of LM8 cell metastatic potential) the results don’t describe the metastatic potential of the two cell lines. Indeed, only in vitro assays were shown that describe proliferation, migration and invasion of the two-cell line. In order to talk about the metastatic potential in vivo work that show metastases formation in a target organ (lung) following inoculation of these cells should be included. The author should change the title to 3.3. Evaluation of LM8 cell malignant phenotype or provide in vivo data if they want to keep the title.

Round 2

Reviewer 1 Report

Thanks for considering my comments and answering my questions. However, the point that you made regarding confirmation with another inhibitor is not reasonable. At least, cell viability needs to be tested with an additional inhibitor as a general rule of thumb.

Reviewer 2 Report

No comments
